# The Shoot Apical Meristem: An Evolutionary Molding of Higher Plants

**DOI:** 10.3390/ijms25031519

**Published:** 2024-01-26

**Authors:** Tania Kean-Galeno, Damar Lopez-Arredondo, Luis Herrera-Estrella

**Affiliations:** 1Institute of Genomics for Crop Abiotic Stress Tolerance, Plant and Soil Science Department, Texas Tech University, Lubbock, TX 79409, USA; tkeangal@ttu.edu (T.K.-G.); damar.lopez-arredondo@ttu.edu (D.L.-A.); 2Unidad de Genómica Avanzada/Langebio, Centro de Investigación y de Estudios Avanzados del Instituto Politécnico Nacional, Irapuato 36821, Mexico

**Keywords:** shoot apical meristem (SAM), SAM conservation, Sc-RNA-seq, streptophyta

## Abstract

The shoot apical meristem (SAM) gives rise to the aerial structure of plants by producing lateral organs and other meristems. The SAM is responsible for plant developmental patterns, thus determining plant morphology and, consequently, many agronomic traits such as the number and size of fruits and flowers and kernel yield. Our current understanding of SAM morphology and regulation is based on studies conducted mainly on some angiosperms, including economically important crops such as maize (*Zea mays*) and rice (*Oryza sativa*), and the model species Arabidopsis (*Arabidopsis thaliana*). However, studies in other plant species from the gymnosperms are scant, making difficult comparative analyses that help us understand SAM regulation in diverse plant species. This limitation prevents deciphering the mechanisms by which evolution gave rise to the multiple plant structures within the plant kingdom and determines the conserved mechanisms involved in SAM maintenance and operation. This review aims to integrate and analyze the current knowledge of SAM evolution by combining the morphological and molecular information recently reported from the plant kingdom.

## 1. Introduction

Plants can maintain indeterminate postembryonic growth by reserving pools of stem cells. These reservoirs are confined to specialized regions called meristems, which are present in various plant structures. The apical meristems (AMs) on the tips of the germinating seedling are responsible for the plant’s primary growth or length. There are two types of AMs: the shoot apical meristem (SAM) and the root apical meristem (RAM), which are responsible for the growth of the aerial part and the root system of plants, respectively. Additionally, plants have lateral meristems, which are responsible for secondary growth that determines the thickness of the plant; intercalary meristems, responsible for growth at the base of the nodes and leaf blades, mainly in monocots; and buds, which usually contain a small mass of meristematic tissue in spermatophytes [1,2,3].

Beyond growth, meristems also play a crucial role in defining plant anatomical patterns. With noticeable differences in their structure, meristems serve as a valuable model for studying plant evolution and understanding the structural and functional aspects of plant development. Primitive plants generally have a simple meristem, usually with one apical cell, whereas late divergent plants have complex multicellular meristems [4]. Despite these important structural variations, meristematic cells share two essential and highly conserved functions: maintaining the meristem integrity, including its stem cell population, and producing cells to support plant growth [5]. This review summarizes current knowledge on meristem structure, function, and evolution. Given the broad scope of this topic and the scarcity of studies dedicated exclusively to studying SAM from an evolutionary perspective, we integrate and synthesize all relevant literature on meristem research. The goal is to provide a comprehensive review that facilitates understanding of SAM evolution. The main questions we aim to address are: (1) Do plant meristems have a single origin, or did multiple independent events give rise to plant meristems? and (2) Are the circuits regulating the maintenance of stem cells conserved during plant evolution? 

This review was prepared by searching all bibliographies on plant meristems in PubMed and Google Scholar databases following the PRISMA methodology (http://prisma-statement.org/PRISMAStatement/default.aspx?AspxAutoDetectCookieSupport=1, accessed on 15 July 2023) for the literature search. The search was filtered using keywords including shoot apical meristem, root apical meristem, plant stem cells, evolution of plant meristems, and single-cell analysis of plant meristems. When multiple papers were found on the same topic, the most complete and updated reports, with the most citations, were selected. We divide this review into four main sections:In the first section, we present essential concepts to understand plant meristems. We introduce the concept of stem cells and compare the stem cell niche of RAM and SAM. We emphasize the importance of plant life cycles, as this concept gives rise to SAM evolution theories.Then, the meristem shape is analyzed throughout phylogeny. This section offers a morphological description of SAM across plant evolution. Given the extensive literature on SAM morphology, it provides perspectives on the differences in SAM between clades.We provide a comprehensive review of the regulatory and maintenance mechanisms in the SAM. This section focuses on the regulatory loops described for angiosperms and the conserved elements across clades.Taking advantage of single-cell transcriptomics to understand SAM, this section delves into the research of single-cell transcriptomics and single-nucleus transcriptomics on SAM. We describe key studies and discuss their findings.

## 2. Evolutionary Origin of the Meristem

### 2.1. The Concept of Stem Cells

Plant stem cells are typically found in meristems, and their defining characteristics include being cells specialized in producing new cells and organs, self-maintenance, slow proliferation with an extended G1 (or G0) cell state, pluripotency (ability to differentiate into multiple cell types), and regenerative potential or the ability to repopulate after damage [6,7,8]. The study of apical meristems (AMs) is particularly intriguing due to their stem cell pool, also known as promeristems, located at the central part of the shoot and the root apex. A continuous differentiating cell flux occurs throughout plant development from stem cells to proliferating cells, finishing with differentiating cell states [9]. Two types of plant cell differentiation exist: terminal and non-terminal. Terminal differentiation implies that a cell can no longer change its fate, while non-terminal differentiation allows a cell to change its fate when exposed to the correct signals, such as normal ontogeny, wounding, or other physiological changes [10]. Cells undergoing non-terminal differentiation appear to retain a stem cell potential [3]. However, the specific conditions activating their differentiation into various cell types remain largely unknown [11]. Standard models for studying plant cell differentiation include embryos, protoplasts, callus, and meristems [11,12,13,14]. These models are valuable as they contain pluripotent or totipotent cells that ultimately give rise to different plant tissues and even regenerate whole plants. Hence, an interesting perspective to understand this phenomenon is to focus on the evolution of pluripotency by comparing the meristems of early and modern diverged land plants, pointing out the common factors of pluripotency through evolution.

### 2.2. SAM and RAM

Extensive research has been conducted to understand RAM evolution, but little is known about the SAM evolutionary process. Since both AMs are structures with similar characteristics, studying the RAM can be a starting point for understanding the SAM. Some common attributes are that both are reservoirs of stem cells, both are promeristems that give rise to five primary meristems (protoderm, ground meristem, procambium, pericycle, and calyptrogen in the root [10]), and they are first determined during embryogenesis [15]. Stem cells in the root apical meristem (RAM) are present and maintained in the quiescent center (QC). In contrast, in the shoot apical meristem (SAM), stem cells are contained in the central zone (CZ) and maintained by the organizing center (OC) [16]. The QC and OC are the signaling centers responsible for stem cell maintenance in both meristems [16] and have been proposed to be functionally equivalent [17,18].

Studies conducted on the model plant *Arabidopsis thaliana* suggest that the organization of the signaling components required for stem cell initiation and maintenance in the RAM and SAM are relatively conserved. Consequently, similar classes of genes have been co-opted as central regulators in both types of meristems [18,19]. However, despite these shared mechanisms, the hormonal function is inverse [20,21]. In the SAM, auxins trigger differentiation, whereas in the RAM, they maintain the stem cell niche and support cell proliferation [22]. On the other hand, in the SAM, cytokinin promotes tissue proliferation of differentiating cells, whereas it promotes cell differentiation in the RAM [23,24]. The OC of the SAM is the site of maximal cytokinin activity, whereas the auxin maximum is in the QC of the RAM [24] (Figure 1). 

Fossil evidence suggests that the RAM evolved at least twice independently rather than having a single origin [25,26]. The first appearance occurred in the lycophytes clade, followed by a second evolutionary event, likely in the ancestor of euphyllophytes (vascular plant non-lycophytes) [27]. It has even been proposed that RAM independently evolved multiple times in lycophytes [28]. Moreover, the similarities between lycophyte and euphyllophyte roots, including indeterminate growth, apical meristem protected by a root cap, root hairs, and a stele covered by a specialized endodermal cell layer, are clear examples of convergent evolution [29,30]. Genomic analyses revealed that limited gene expansion occurred at the divergence between the lycophyte and euphyllophyte clades [31]. Consequently, all the similarities produced by convergent evolution did not require extra gene families, suggesting that the rewiring of existing genetic programs was sufficient to generate multiple independent emergences of the RAM [32].
Figure 1Diagram of SAM and RAM in *Arabidopsis thaliana*. In the SAM, a stem cell pool is located in the Central Zone (CZ, purple) above the organizing center (OC, yellow), which expresses the WUSCHEL transcription factor (TF) (yellow). Cells that pass the boundary defined by CLAVATA (CLV) function start differentiation, establishing the organ founder cell population. The RAM consists of a pool of quiescent cells (QC, cyan) surrounded by a pool of initial cells (the consideration of initial cells as stem cells depends on the author). This regulatory model highlights the complex interplay of phytohormones and TFs in the WOX domain. Arrows indicate activation and barred lines indicate inhibition. The dashed arrows beside auxin (AUX) and cytokinin (CK) indicate the hormone flow direction. AUX produced in the SAM and young leaves are basipetally transported through the stem by the polar auxin transport (PAT) stream toward the RAM. CK biosynthesis genes are expressed in the RAM differentiation zone and acropetally transported. CZ, central zone; PZ, peripheral zone; LP, lateral primordium [33,34].
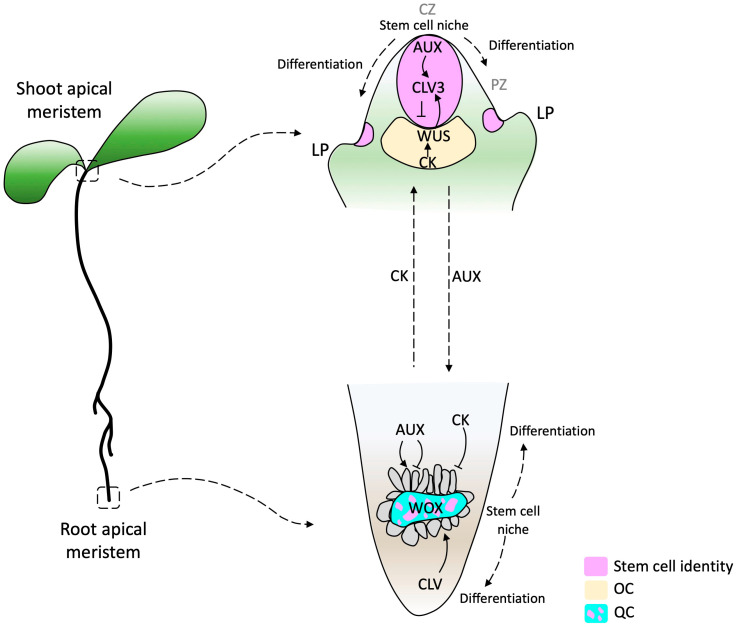


### 2.3. Apical Meristem of Gametophyte and Sporophyte

The plant life cycle comprises two alternating phases: the haploid or gametophyte phase and the diploid or sporophyte phase. The diploid phase concludes with meiosis, while the haploid phase ends with the generation and union of gametes [35]. It is remarkable that, as land plants diversified, life cycle generation dominance twisted, and the sporophyte generation became dominant and increasingly specialized, whereas the gametophyte generation became reduced [36,37]. This trend is reflected in SAMs. Non-seed plants have simple meristems, sometimes with a single apical cell, to sustain their growth. In contrast, seed plants’ sporophytes develop complex multicellular meristems to support their growth and specialization, while their gametophytes lack meristems [38,39] (Figure 2). As both phases have meristems, it is reasonable to discuss the origin of meristems in terms of the plant life cycle. However, we must mention that we are focusing on SAM, specifying apical meristems of the vegetative growth (independent of the presence of leaves). We include some meristems without or with reduced vegetative growth (as in bryophytes) [40] as they are attractive for understanding the evolution of SAM.

Two hypotheses have been proposed that explain the appearance of the alternation cycle [41]. First, the homologous hypothesis describes the ancestor of land plants with an originally haplontic life cycle that gives rise to a sporophyte through delay in meiosis, resulting in the intercalation of a new diploid organism [37,42]. This assumption outlines the gametophytic SAM as the ancestor of the sporophytic SAM. The second hypothesis is the antithetic hypothesis, which proposes that sporophytic SAM is independent of gametophytic SAM. According to this hypothesis, the sporophytic SAM evolved de novo by intercalating a system between embryonic and reproductive growth [37,43]. Current evidence based on embryophytes and fossil records is more compatible with the antithetic hypothesis than the homologous hypothesis [44,45]. Hypotheses derived from the antithetic hypothesis include (1) the embryophyte SAM originated from the transient seta meristem (sporophyte) of a bryophyte, (2) the hornwort sporophyte meristem is the possible ancestor of the SAM, and (3) the SAM in vascular plants arose de novo from a bryophyte ancestor [46]. 

Overall, evidence suggests that gametophyte and sporophyte SAM have an independent origin. However, because plants have retained a common set of genetic components since they diverged from algae, they are likely to share homologous regulatory loops and mechanisms in regulating their SAM. In fact, moss and fern gametophyte SAMs share key transcription factors (TFs) with angiosperm sporophyte SAMs [47,48], which suggests the existence of fundamental mechanisms involved in stem cell regulation across land plants.

## 3. Meristem Shape throughout Phylogeny

A traditional approach to studying meristems considers their morphology and focuses on their correlation to plant growth patterns. Cell division patterns and apical cell symmetry can outline plant morphology [49]. For example, in bryophytes, taxa with tetrahedrical apical cells tend to have leafy morphologies, while taxa with hemidiscoid apical cells tend to be thalloid [50]. The study of apical cells applies mainly to SAMs with a single apical cell found in macroalgae, bryophytes, lycophytes, and ferns. The simplest apical cell is the dome-shaped cell in macroalgae (Charales) and moss gametophytes, allowing one-dimensional (1D) growth. However, land plant clades such as lycophytes and ferns present a tetrahedral apical cell enabling three-dimensional (3D) growth [51,52] (Figure 3). It can be hypothesized that the emergence of apical cells capable of dividing following multiple planes resulted in drastic morphological innovations, ultimately allowing the colonization of land by plants.

### 3.1. Algae 

The evolution from algae to plants is closely tied to the evolution of the body plan of plants. The Charophycean green algae, the closest relative of the extant land plants [63,64], provides an evolutionary example of the transition from unicellular to multicellularity. Some Chlorophycean algae are still at the midpoint of this transition [65]. However, the presence of a SAM is limited to multicellular organisms with organized tissues. 

Among Charophyceans, apical growth has been demonstrated only in the order Charales, which have a single apical cell per shoot [64]. The dome-shaped apical cells of Charales have one cutting face opposite the apex of the surface, and this cutting plane is parallel to the orientation of the last cell plate. As a result, they present 1D growth, forming cell chains resembling filaments or mats [51,53]. Another charophyte, the Coleochaetalean algae, typically exhibits a marginal meristem where cells can divide into two possible planes, leading to the formation of radially symmetrical filamentous or monochromatic thalli that resemble hornwort gametophytes [66].

### 3.2. Bryophytes

Bryophytes comprise a taxonomic group consisting of three clades of non-vascular land plants: mosses, liverworts, and hornworts [67]. It is traditionally considered that land plant traits primarily evolved in bryophytes. Liverworts were the earliest divergent clade of land plants, whereas mosses and hornworts are sister groups. However, there is only partial acceptance of the internal topology of this clade despite evidence of its monophyly [68,69]. It has been proposed that the sporophyte SAM was derived from a sterilized bryophyte sporangium, but this hypothesis remains untestable due to a lack of homologies or fossil records [70,71].

The moss sporophyte has three types of cells recognized as meristems: the apical cell, active in the early embryo; the central seta cell that maintains the seta’s growth; and a single apical cell of the sporophytic tissue [50,54,72]. This anatomy is similar to that of liverworts, except that, in mosses, the seta arises from a transient meristem developing in the middle of the spindle-shaped embryo. In contrast, hornworts lack a seta and, consequently, a seta meristem on their sporophyte. The hornwort sporophyte is composed of a foot and a sporangial axis, growing from a basal meristem that remains active throughout the sporophyte’s life [37,73]; it possesses a unifacial and multicellular basal meristem, called the basal meristem, which provides upwards indeterminate growth. This type of “indeterminate” growth is unprecedented to any fossil or living plant and is similar to the indeterminate growth of sporophytes in tracheophytes. However, the resemblance to the angiosperm SAM is considered unrelated or superficial [74,75]. The hornwort’s indeterminate growth contrasts with the limited proliferative growth of sporophytes in mosses and liverworts. 

Bryophytes have an extended gametophytic phase during their life cycle. The gametophyte of bryophytes presents a long-lasting single-celled SAM composed of an apical cell (AC). Among bryophytes, mosses exhibit a singular development compared to hornworts and liverworts [38]. After spore germination, mosses develop filamentous tissues called protonema, consisting of two substages. In the early germination stage, the protonema is composed of chloronemata filaments. The chloronemata exhibits 1D growth during filament extension until an apical stem cell transitions into a caulonemal apical stem cell. The caulonema consists of an extended 2D filamentous stage. Both substages comprise a protonema apical cell. Lastly, moss initiates 3D growth and leafy shoot buds called gametophores with a single tetrahedral apical cell [53]. In contrast, hornworts and liverworts do not possess a protonemal stage; after spore germination, they grow directly into thalli or “leafy” forms [54,76]. Liverworts develop from apical cells located in an invaginated notch. There are four different types of apical cells found in this group. These include a tetrahedral cell with three cutting faces, a wedge-shaped cell with two lateral, one dorsal, and one ventral cutting face, a lenticular or lens-shaped cell with two lateral cutting faces, and a hemidiscoid cell with two lateral and one posterior cutting face [54]. On the other hand, hornwort gametophytes usually present wedge-shaped apical cells with four cutting faces located in notches around the thallus [77].

### 3.3. Tracheophytes

Numerous efforts have been made to categorize the SAMs of tracheophytes, primarily through their histological and cytological features. It is generally accepted that there are three types of SAMs. This classification is based on the division plane within the SAM cells and the location and number of initial cells [78,79]. Initial cells divide to maintain the meristem as a continuing source of new cells. The three types of tracheophyte SAMs are (1) monoplex type, usually capped by one apical initial cell, also known as apical cell, that contributes to stem growth [80]. This SAM type is found on some lycophytes and ferns; (2) simplex type, whose configuration contains multiple initial cells within a single zone. These are common in gymnosperms [80]; (3) duplex meristem type that incorporates initial cells distributed across at least two layers [61,80]. These are prevalent in angiosperms.

The indeterminate meristem of tracheophytes is a crucial innovation in plant evolution, which pushed the sporophyte life cycle as the dominant phase in the plant’s life cycle [81]. Lycophytes comprise a taxonomic group representing a small portion of vegetal diversity. This clade consists of three living orders: Lycopodiales, Isoëtales, and Selaginellales. However, there are many other extinct orders [82]. Lycophytes display considerable variation in the organization of their meristems, ranging from the monoplex meristem with one or two apical cells found in Selaginellales [58,83] to the simplex meristems found in Lycopodiales and Isoëtales [61]. Monilophytes (ferns) usually have a meristem composed of an apical initial cell and its surrounding cells; this multicellular structure has also been proposed for Selaginella [47,84]. Apical cells of Lycophytes and Monilophytes (ferns) are considered convergent structures [47,85].

SAMs in seed plants tend to be larger and more complex compared to other land plants. They consist of multiple initial cells and distinct functional layers. The outer layer (L1), also known as tunica, gives rise to the epidermis, and L2 in maize, or L2 and L3 (corpus) in Arabidopsis, are the internal layers [86,87,88]. Gymnosperm SAMs lack tunica, but the cells in the upper portion of the gymnosperm SAM form a lens-shaped zone with cytological features similar to those of angiosperms. Remarkably, there are gymnosperm genera with an apparent or discrete outer L1 layer, such as Gnetum and Ephedra [61].

SAMs exhibit zonation patterns in gymnosperms and angiosperms, often called cellular domains, which reflect cell division or activity variations, regardless of the functional layers. Different zones of the meristem possess different functions. Typically, there is a central zone (CZ) surrounded by a peripheral zone (PZ) on the flanks and a rib zone (RZ) below the CZ. Cells within the CZ self-renew and replenish the surrounding (PZ) and subtending (RZ). Cells within the PZ and RZ proliferate more rapidly than in the CZ and ultimately differentiate to produce lateral organs and ground tissue [60,89]. 

## 4. Shoot Apical Meristem Regulation and Maintenance

The phenomena of branch development along the primary axis of plants, known as apical dominance, have been studied since the 1930s [90,91,92]. Over nine decades of research, a general concept emerged: the phytohormone auxin produced by the shoot tip is transported in a basipetal fashion by the polar transport stream, which inhibits axillary bud outgrowth [90,93]. Although this concept has been observed in angiosperms and gymnosperms [2,94,95,96], it has not been validated in other growth systems as dichotomous branching. 

Auxin was the first plant regulator explored in SAM activity, and much has been learned about its signaling pathways in different organs. However, particular attention has been given to its interaction with cytokinin, another plant hormone. Auxins and cytokinins act synergistically or antagonistically to control SAM organization, formation, and maintenance [97]. For example, in Arabidopsis, cytokinins promote cell expansion, increasing SAM size, while auxin indirectly promotes differentiation through multiple mechanisms [98]. Despite being the most extensively studied regulators of plant development across plants, our understanding of the crosstalk between these hormones is still limited to a few plant models.

The apical dominance mechanism explains the classical observations; however, in addition to auxins and cytokinins, strigolactones (SLs) constitute a new class of phytohormones related to this mechanism. SLs were not considered in the classic literature because they were not introduced until the 2000s [99,100] and were soon linked to multiple developmental processes, such as shoot development [101,102,103,104]. Specifically, SLs play a crucial role in repressing bud outgrowth in monocots and in responses to environmental factors [102,105,106]. Recent studies suggest that the inhibition of bud outgrowth in apical dominance is attributed to the modulation of apically derived auxin flux by cytokinin and SLs [101,102].

### 4.1. The Regulatory Model: Angiosperms

With advances in molecular biology, SAM research has expanded from physiological aspects to the regulatory molecular mechanism. We understand that phytohormones and TFs cooperate to balance meristem maintenance and organ production. The canonical model of SAM regulation is based on the plant model *Arabidopsis thaliana*, and stands around the key genes CLAVATA (CLV, encoding ligand peptides) and WUSCHEL (WUS, encoding a homeobox TF) [107]. This regulatory loop also involves SHOOT MERISTEMLESS (STM), a KNOTTED1-LIKE HOMEOBOX (KNOX) TF [108,109]. The CLV gene family comprises the CLV peptides CLV3, CLV2, and CLV1. In this regulatory loop, WUS promotes stem cell identity and is regulated by CVL and STM. In addition, CLV favors organ initiation, and STM prevents the incorporation of central meristem cells into organ primordia [107,110] (Figure 4). 

Other genes, such as the plant-specific GRAS TFs and HAIRY MERISTEM (HAM1–HAM4), interact with WUS/WOX5 (another member of the WUS family) [111]. A HAM concentration gradient modulates the WUS–CLVs interaction, promoting zonation of the SAM. In addition, members of the NAC group of leucine-rich TFs, including CUP-SHAPED COTYLEDON1 (CUC1), CUC2, and CUC3 [112], repeat receptor-like kinase genes (LRR-RLK), REVOLUTA (REV, a homeodomain TF), and the APETALA 2 (AP2) TF family, have been shown to have critical functions in the SAM [113,114].
Figure 4SAM signaling pathways in various species. This figure displays genes and factors, such as cytokinin (CK), with established genetic and biochemical interactions. Arrows indicate positive regulation, while barred lines represent negative regulation. Question marks denote unidentified receptors. Marchantia does not have a well-described loop to maintain meristem regulation. However, it has been reported that JINGASA (JIN) acts downstream of CLV3/ESR-related (CLE) peptide signaling and controls stem cell behavior in the gametophyte [115]. The structure of the moss meristem was originally reported by Hata and Kyozuka [38], and those of maize and Arabidopsis by Fletcher [116].
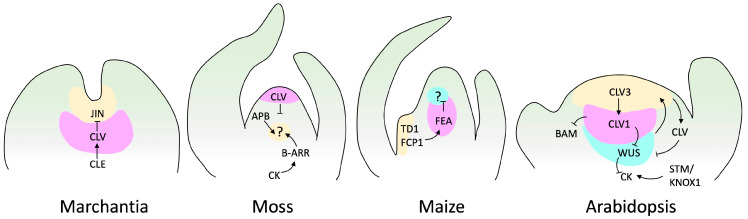


### 4.2. SAM TFs Conserved throughout Evolution

#### 4.2.1. KNOX TFs

The KNOX genes belong to a large family of TFs called homeobox. They are involved in morphogenesis in all major eukaryotic lineages. In plants, members such as KN1, STM, and KNAT are directly associated with SAM maintenance [64]. Conservation of the KNOX gene family can be traced back to the plant’s ancestors, including Chlorophytes and Charophycean algae [64]. It has been suggested that KNOX transcription factors are part of the genetic toolkit that enabled the development of multicellularity, cell differentiation, and increased SAM complexity in seed plants [117]. Their functional roles are diverse across the phylogeny, including activation of the diploid phase, sporophyte and spore formation, meristem maintenance, and organogenesis [118,119]. In organisms such as *Chlamydomonas reinhardtii*, KNOX, and BELL TFs are inherited by gametes of the opposite mating types and heterodimerize in zygotes to activate diploid development [119,120]. Similarly, in Marchantia *polymorpha*, the expression in the gametophyte of KNOX and BELL is required to initiate zygotic development, while in *Physcomitrella patens*, KNOX expression is associated with sporophyte development and meristem regulation rather than the gametophyte [73]. In Lycophytes such as *Sellaginella* and spermatophytes, KNOX genes have been associated with cell proliferation and meristem maintenance [121,122]. Unfortunately, until now, there has not been a clear pattern of KNOX function across the phylogeny. However, duplication events followed by a sub-functionalization within each lineage appear to explain the presence of paralogs specialized for several developmental functions [64].

#### 4.2.2. MADS TFs

The MADS-box gene family encodes TFs with a DNA-binding MADS domain, which was named after the proteins MINICHROMOSOME MAINTENANCE 1, AGAMOUS, DEFICIENS, and SERUM RESPONSE FACTOR (SRF). They are classified into two major classes based on their structure and phylogeny, Type I and Type II [123]. Type II classic MADS-box genes have been particularly well studied, as many have roles in determining floral organs. Type II MADS-box genes further diverged into two groups: MIKCC and MIKC* a [124]. Phylogenetic analyses suggest that algae MIKCC MADS-box genes could be considered the ancestral MIKCC before their divergence into the MIKC* and MIKCc clades. In Charophycean algae, MIKCC genes play a role in gamete differentiation [125].

In non-seed plants, MIKCCs have roles in gametophytic and sporophytic generations and contribute to the development of vegetative and reproductive structures [126,127,128]. However, in seed plants, their function is primarily linked to sporophyte development and the determination of floral organs [129,130]. On the other hand, MIKC* genes of non-seed and seed plants have a conserved role during gametophyte development [128,131]. It has been hypothesized that the function of MADS-box genes became restricted to specific plant organs after duplication events coinciding with the diversification of seed plants. Before that, MADS-box genes had multiple roles throughout plant development [132]. 

#### 4.2.3. AP2/ERF TFs

The plant-specific APETALA 2/ethylene-responsive factor (AP2/ERF) family is characterized by the AP2 DNA-binding domain [133]. *AP2/ERF* genes are divided into classes based on the number of AP2 domains present. *ERF*-like genes contain one AP2 domain, while *AP2*-like genes contain two AP2 domains [134]. AP2-like genes can be further divided into the clades euANT, basalANT, and euAP2 according to the amino acid sequence of the double AP2 domain and the nuclear localization sequence [135]. In the model plant *Arabidopsis thaliana* and other angiosperms, all AP2-like clades play key roles in developmental processes and SAM regulation [117,136,137]. 

The AP2/ERF family is involved in diverse roles through plant evolution, although there is no experimental evidence of AP2-like function and no ANT sequences in algae. Phylogenetic analysis showed that an AP2 domain (AP2-R1 AA insertion) of the microalgae *C. reinhardtii* and *Chlorokybus atmophyticus* form a sister clade to the major clade of the AP2-like sequences of plants. However, they are distinct from euANTgene sequences, forming their own clade [138]. This suggests that *C. reinhardtii* and *C. atmophyticus* AP2 sequences could represent the putative ancestral sequence of the ANT group. In contrast, the AP2-like genes of multicellular algae, such as *Mesotaenium caldariorum* and *Klebsormidium nitens*, form a clade with land plants. It has been suggested that the ancestor of embryophytes may have a preANT-like gene which gave rise to the land plant exclusive basalANT and euANT lineages [134,137]. However, the diverse functions of this gene family increased following the plant’s evolutionary novelties; for example, in ferns, the expression of ANT has been reported in young sporangia, gametes, and spores, in gymnosperms in the ovule and during seed development. In angiosperms, they are involved in the meristem, flower organ, and fruit development [137,139,140].

## 5. Taking Advantage of Single-Cell Transcriptomics to Understand SAM

Studies on SAM transcriptional pathways have evolved from early techniques, such as in situ hybridization and microdissection to more high-throughput methods, such as single-cell RNA-sequencing (sc-RNA-seq) and single-nucleus RNA-sequencing (sn-RNA-seq). In situ hybridization is used to map specific genes, making it difficult to obtain a comprehensive view of the transcriptome. Microdissection involves dissecting tissue pieces, from which RNA is extracted for analysis. Unfortunately, microdissection studies are susceptible to RNA degradation and low yields, potentially leading to low-quality data [140,141]. scRNA-seq and snRNA-seq (sc/snRNA-seq) are techniques that avoid these problems. sc/snRNA-seq relies on microfluidics (10× Genomics) to capture single cells or nuclei and obtain an RNA sequencing library for each cell or nuclei. After sequencing, the transcriptomes of individual cells are used to group cells into clusters, and clusters are assigned to cell types using previously described markers from fluorescent protein reporters or in situ hybridization [142]. 

sc/snRNA-seq simultaneously allows the study of cellular dynamics, including the cell cycle, cell differentiation, and gene regulatory network construction [143]. This is particularly useful in studying meristem development, as it allows the identification of stem cells and their exploration during transitional states up to their differentiation [144]. SAM transcriptomic landscapes are a turning point in plant development research. Sc/snRNA-seq offers the possibility to study division and differentiation at cell resolution, quantifying the contribution of individual cells during the formation of new aerial structures, a task that was impossible before this technology [15]. Currently, three SAM scRNA-seq studies in Arabidopsis [145], one in maize [146], and one in pea [147], and two SAM snRNA-seq studies, one in tomato [148] and one in *Populus* [149], are available (Table 1).

The transcriptomic landscape of the *Arabidopsis* SAM is an invaluable reference due to its status as the most extensively studied plant model species and has led to numerous resources for single-cell analysis. Three main conclusions can be drawn from *Arabidopsis* SAM scRNA-seq research. First, despite the wealth of available markers, there remain undefined groups of cells, hinting at the potential existence of new cell types. Second, this research also explores the dynamics of transcriptomic changes during temporal processes, enabling the assignment of cells into distinct cell cycle phases, which revealed a cell cycle continuum in meristematic tissues, suggesting that subtle variations in cell division duration and patterns play a role in the formation of different cell types. Third, the overlap of single-cell RNA sequencing data from *Arabidopsis* RAM and SAM uncovered previously unrecognized similarities between shoot and root apex epidermis and vascular tissues, adding to our understanding of plant development [145].

Conde and colleagues [149] developed a SAM cross-species analysis between Populus and *Arabidopsis* at single-cell resolution. These results provide novel insights into the conservation of transcriptional programs during primary vasculature formation and yield valuable information about xylem and wood formation. The tomato SAM transcriptomic landscape described by Tian and colleagues [148] showed the enrichment of *Arabidopsis* homologous genes associated with the cell-specific domains, such as meristem, epidermis, mesophyll, and vasculature. Furthermore, Tian and colleagues [148] used a single-cell approach to infer gene regulatory networks, facilitating the identification of key regulators.

For the first time, Chen and colleagues [127] used sc-RNA-seq technology on pea SAMs to understand plant growth under nutrient stress, specifically boron (B) deficiency. Their study links the progression of SAM development with the upregulation of genes encoding histones and chromatin assembly and remodeling proteins under B deficiency conditions. The expression of these genes was suppressed, suggesting a reorganization of chromatin during SAM development and a possibly impaired SAM activity under B deficiency. These results emphasize the importance of considering cell type-specific stress responses, which may be key to unraveling complex biological processes [147]. 

Satterlee and colleagues [146] obtained a transcriptional landscape of the maize SAM. Maize belongs to the monocot clade, while *Arabidopsis*, *Populus*, tomato, and pea are eudicots, and one would expect to observe more differences between the transcriptional profiles of monocots vs. eudicots. Based on their scRNA-seq data, Satterlee and colleagues [146] did not find evidence of a WUS-CLV3 regulatory loop as in other SAM transcriptional landscapes. The *Arabidopsis* SAM exhibits a repression mechanism of the stem cell-promoting transcription factor WUS, which relies on the peptide CLV3 and its receptor CLV1, expressed in cells in the CZ. WUS is crucial in maintaining the stem cell fate in SAM and floral meristems (FM). However, there is no clear regulatory loop like that in *Arabidopsis*.

Maize SAM has several Leucine-Rich-Repeat (LRR) receptors and their peptide ligands that could be involved in stem cell control [150,151], but the identity of the TF(s) that promote the stem cell fate remains unclear. WUS1 has been considered an essential meristem size regulator in maize [113]. Its expression has been detected in the inflorescence meristem (IM) and FM [152]. In addition, in situ hybridization localizes WUS in the SAM. However, in contrast to *Arabidopsis*, where WUS expression is restricted to the organizing center (OC), in maize, it is detected in peripheral cell layers at the primordium height [153]. It is noteworthy that in maize, SAM sc-RNA-seq failed to detect WUS or any functionally homologous genes [146]. Laureyns and colleagues [154] developed an in situ sequencing (ISS) on maize SAM meristem. They probed the expression of 90 genes and could not obtain a clear expression pattern for 15 of them, including ZmWUS1 and ZmWUS2.

Several challenges deriving from the protoplasting technique could hinder maize SAM heterogeneity due to issues arising from this process. Protoplasting has been linked with the activation of ectopic gene expression, bias on proportions of cell types [13,145], and noise gene expression related to the activation of regeneration and cell division mechanisms after protoplast isolation [113,148]. Therefore, it is possible that WUS-containing cells were affected during protoplast isolation, or the sensitivity of the in-situ sequencing (ISS) could not capture the low level of WUS expression. Considerations on this phenomenon are that the expression of WUS within the maize meristem exhibits variation based on its developmental stage, or there is no WUS-CLV3 regulatory loop in maize SAM under normal conditions. Alternatively, WUS could function as a mobile TF [152], raising the possibility that in the maize SAM, WUS has a different expression pattern than in other plants. Remarkably, this example in angiosperms between monocots and dicots exposes the possibility of different mechanisms in SAM regulation. Therefore, thinking about the diversity in regulatory circuits operating in the SAM regulation in different clades is exciting. 

Using sc/snRNA-seq to study the SAM in non-model plant species has additional challenges. Sc/snRNA-seq relies on previous knowledge of cell type-specific markers to assign cell clusters to cell types. Such markers are rare to non-existent in non-model plant species. Spatial transcriptomics has emerged as an alternative to overcome this limitation, although it has drawbacks. The spatial transcriptomics technique enables creating in situ libraries, thus obtaining transcriptomic landscapes from tissue sections, avoiding the need to cluster and identify each cell population [155]. Giacomello and colleagues [156] established the basis of the spatial transcriptomics method for SAMs of different plant species [15,157]. The possibility of matching the anatomical and molecular information and accelerating gene and tissue discovery is a valuable resource for future functional genomics studies. Additionally, technologies such as spatial transcriptomics and snRNA-seq will facilitate the study of non-model species as it becomes unnecessary to develop protocols for protoplasting or extensive background on marker genes for particular species. These technologies alleviate the technical difficulties of working on SAMs, as seen in the examples above. However, while there are remarkable advantages to using sc/snRNA-seq for discovering genes and TFs, functional assessment of the regulation dynamics can only be achieved through genetic and genomic approaches.

## 6. Perspectives

We have embarked on the synthesis of SAM from an evolutionary perspective. However, broadening our view to encompass an evolutionary perspective has inherent limitations. While this approach has been a subject of discussion for some time, early research was restricted by microscopy techniques, resulting in a substantial body of literature that primarily focused on descriptive studies and qualitative data, often employing non-standardized characters for comparison. Nevertheless, refined microscopy techniques are now available and facilitate the morphological comparison of meristems between species [24,68]. In general, techniques for studying meristems at various levels have only recently been developed. In addition to these limitations associated with evolutionary perspectives, the lack of models spanning phylogeny is notable. Consequently, in silico, in situ, and in vitro studies are only sometimes available for particular clades, creating significant gaps in our understanding of the evolutionary processes.

The study of the SAM is a topic that has captivated humanity historically. Initially, based on anatomical approaches, the prevailing evolutive notion was that the simple meristems composed of one or a few cells were the most primitive, with increased complexity during plant evolution. However, our current understanding suggests that the SAM has multiple origins and that meristems with an apical cell may reappear in vascular clades from ancestors with multicellular meristems [4]. However, further evidence is needed to confirm and specify the multiple origins of SAM. Among all groups of plants, angiosperms are the most studied. However, research into the conservation of SAM regulatory pathways is still in its early stages.

Angiosperm meristems, mainly dicot meristems, have been extensively studied, and there is an increasing interest in deciphering whether SAM regulatory pathways are conserved among diverse groups of plants. Evidence suggests that monocot and dicot meristems could exhibit distinct regulatory loops. However, if such diversity exists within angiosperms, one might wonder about its prevalence in other plant groups. For instance, does research on meristem regulation in bryophytes and lycophytes need to commence without assuming similar regulatory mechanisms?

Significant progress has been made in understanding the maize SAM through single-cell and spatial transcriptomics. However, developing this technique in non-model species remains challenging mainly for three reasons. First, protocols for sc/snRNA-seq in plants are species-dependent. Second, developing these techniques on meristems with only one apical cell is challenging as the cell recovery on the overall efficiency of current scRNA-seq protocols can vary between <1% to >60% across cells, depending on the method used [158]. Therefore, ensuring the representation of a sufficient percentage of apical cells would necessitate adept technical skills. Third, as sc/snRNA-seq analyses rely on gene markers to identify clusters, the poor genomic information available for non-model species will limit their significance. This will also be impacted by the lack of information regarding cellular domains of the SAM and gene expression and regulatory pathways involved in regulation. Therefore, all this information must be generated in advance.

SAM is typically studied under standard growth conditions. However, environmental stresses can significantly impact plant growth, highlighting the crucial link between SAM and stress responses. Investigating this connection may reveal strategies that allow plants to adapt to adverse environments. An interesting perspective, following the sc-RNA-seq study on peas [127], is to extend the research to other legumes, such as the common bean (*Phaseolus vulgaris*), an important crop that is known to be susceptible to drought stress [159]. By employing sn/sc-RNA-seq to analyze SAM bean under drought stress at a cellular level, it is possible to explore the involvement of key TFs in its drought stress response, for example, DREB2 (Dehydration-Responsive Element-Binding Protein 2), a member of the AP2/ERF family, which has been described to play a fundamental role in drought responses [160].

Comparative studies between multiple species have the potential to shed light on the regulatory pathways of the SAM, help clarify its origin, and address related questions. For instance, are there other regulatory cycles besides the CLV-WUS? Do they act in a concerted way or independently? Or are they species-specific? That would help explore the presence of conserved or non-conserved regulatory loops in monocots and other plant groups. Furthermore, it will be essential to investigate whether the expansion of SAM regulatory genes relates to plant morphological traits across plant phylogeny. Another exciting perspective is the study of stress responses at the SAM level to explain early responses to anatomical alterations and to search for conserved mechanisms that promote and maintain cell dedifferentiation and proliferation, which would greatly impact diverse research fields.

With the advent of new technological alternatives, it is now possible to trace the conservation of regulatory pathways and loops, i.e., those involved in stem cell proliferation and maintenance. Additionally, these technologies enable the identification of key genes responsible for specific traits. However, technologies like sc/snRNA-seq analysis are recent in the plant field. Their application in uncovering key genes on stress response has begun to shed light on potential applications in the crop industry and plant improvement.

## Figures and Tables

**Figure 2 ijms-25-01519-f002:**
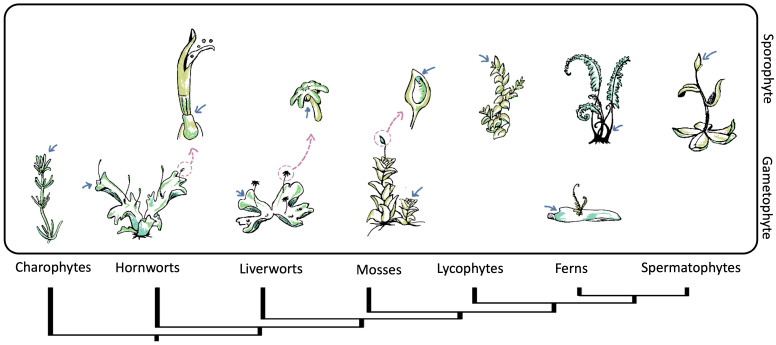
Phylogeny of streptophytes and schematic representation of the shoot apical meristem (SAM) location throughout gametophytic and sporophytic life stages. The pink arrows point to the position of the meristems, while the blue arrows indicate a close-up of specific meristematic structures. Within bryophytes and charophytes, the gametophyte is the dominant life state with vegetative growth from SAMs. Meanwhile in vascular plants, vegetative growth from SAMs occurs in a sporophyte state. The gametophytes of lycophytes and spermatophytes are not presented.

**Figure 3 ijms-25-01519-f003:**
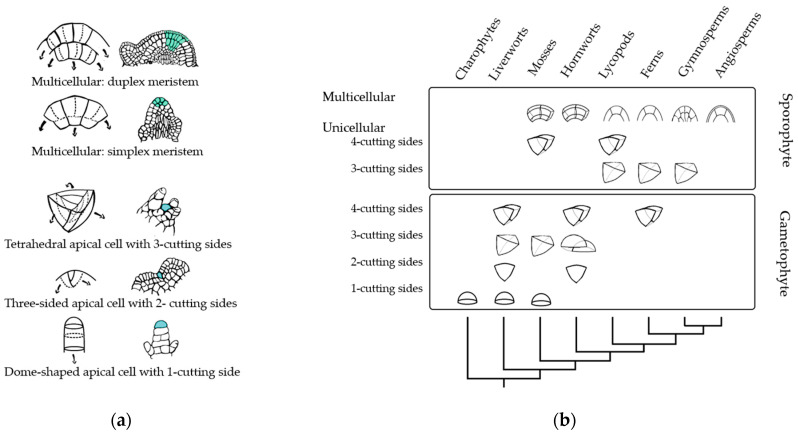
Schematic representation of the organization of different types of apical meristems (AMs) from various taxonomic groups. (**a**) Representation of meristem shapes and division plane of meristematic cells. From top to bottom: cross-sections of duplex and simplex meristems where divisions of the initial cells can occur anticlinally to provide length or periclinally to increase breadth. Meristems with one apical cell: a tetrahedral apical cell with three cutting sides, a three-sided cell with two cutting sides, and a dome-shaped apical cell with one cutting side. Dashed lines represent division planes, and arrows represent the directions in which the cell’s progeny is displaced during growth [34,49]. Blue cells represent stem cells, green shadowed area represents the meristematic cell or meristematic zone. (**b**) Schematic representation of streptophyte phylogeny based on AM anatomy through gametophyte and sporophyte life stages. Within the charophycean algae, the apical meristem is a single cell with one cutting face [51,53]. Within bryophytes, a variety of apical meristems are found, including unicellular meristems with one, two, three, and four cutting sides, as well as simple multicellular meristems consisting of two cells with eight cutting sides [50,54]. Pteridophytes and lycophytes exhibit a wide variety of SAMs; they can have one apical cell with two or three sides, a proliferative zone, and more complex multicellular meristems [55,56,57,58,59]. Large multicellular meristems occur within the seed plants. This figure shows only some common AM shapes described, while multiple shapes can occur between different species [56,60,61]. Panel (**b**) adapted from Niklas and colleagues [62].

**Table 1 ijms-25-01519-t001:** Summary of snRNA-seq and snRNA-seq SAM studies regarding SAM.

Scientific Name	Clade	Technology	Validation	Focus	Reference
*Arabidopsis thaliana*	Dicot	scRNA-seq	GFP-reporter lines of identified genes (At2g38300, At4g34970 (ADF9), At4g11290, and At1g13650).	Construction of developmental trajectories of several tissues and Integrative.	[145]
*Zea mays*	Monocot	scRNA-seq	In situ hybridization of KN1-overexpressor lines.	Analysis of maize SAM stem cell function and cell-fate acquisition.	[146]
*Pisum sativum*	Dicot	scRNA-seq	Laser capture microdissection to identify marker genes.	Cell type-specific responses to boron deficiency.	[147]
*Solanum lycopersicum*	Dicot	snRNA-seq	Homologous markers identification.	Construction of developmental trajectories of several tissues.	[148]
*Populus trichocarpa*	Dicot	snRNA-seq	RNA in situ hybridization and GUS-reporter lines.	Comparisons on Phloem and xylem development between a woody plant and annual herb.	[149]

## Data Availability

No new data were created in this study.

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
