# Peer review of "The Shoot Apical Meristem: An Evolutionary Molding of Higher Plants"

_ijms, 2024, doi:10.3390/ijms25031519_

Round 1

Reviewer 1 Report

Comments and Suggestions for Authors

This is a wonderful compilation of research done on plant meristems beyond what is known from common model and crop plants. I very much enjoyed reading it and only have a few comments and corrections (mostly textual):

Line 85: Please note that (in agreement with Figure 1) the stem cells are contained in the central zone (CZ: purple area) not the OC in the SAM – the zones barely overlap. In contrast, the QC is in the center of the stem cell niche in the RAM. Please rephrase. (same problem in line 449: “stem-cell OC”)

Lines 125 to 132: I don’t understand. How is the reduction of the gametophyte in more advanced land plants connected to the presences of single-celled SAMs in both gametophytes and sporophytes of non-seed plants.

Line 474: What is “a canonical OC”?

Minor:

Line 117: Please revise. Not all dotted (or rather dashed) arrows indicate auxin and cytokinin flow in this figure.

Line 122: Please replace “metistem” with “meristem”

Line 127: Please replace semicolon with full stop in this line.

Line 150: Please correct – “… which suggests the existence of fundamental mechanisms…”

Figure 2: This is a beautiful figure; but please reconsider the use of light blue and light pink for the arrows, as these will not visible. Please align the labels at above the box.

Line 176: Please replace “Tetrahedral” with “tedrahedral”

Line 278: Please replace “behind the CZ” with “below the CZ”

Line 317: Please capitalize “Apetala 2”

Line 344: full stop missing after sentence.

Line 379: Please replace full stop with comma

Lines 493/494: something is off with this sentence. Please rephrase.

Author Response

Reviewer 1 

This is a wonderful compilation of research done on plant meristems beyond what is known from common model and crop plants. I very much enjoyed reading it and only have a few comments and corrections (mostly textual):

 Response: We thank this Reviewer for his/her opinion on our review and the important revisions suggested. The text has been edited accordingly in this revised version of the manuscript, and our responses and corrections can be read below.

Line 85: Please note that (in agreement with Figure 1) the stem cells are contained in the central zone (CZ: purple area) not the OC in the SAM – the zones barely overlap. In contrast, the QC is in the center of the stem cell niche in the RAM. Please rephrase. (same problem in line 449: "stem-cell OC")

Response: We thank this Reviewer for pointing this out. Based on these recommendations, the revised version of our manuscript has been edited accordingly. The said paragraphs now read as follows:

Paragraph starting in new Line 105:

“Stem cells in the root apical meristem (RAM) are present and maintained in the quiescent center (QC). In contrast, in the shoot apical meristem (SAM), stem cells are contained in the Central Zone (CZ) and maintained by the organizing center (OC [12].”

Paragraph starting in new Line 696:

The Arabidopsis SAM exhibits a repression mechanism of the stem-cell-promoting transcription factor WUS, which relies on the peptide CLV3 and its receptor CLV1, expressed in cells in the CZ. WUS is crucial in maintaining the stem-cell fate in SAM and floral meristems (FM). However, there is no clear regulatory loop in maize SAM like that in Arabidopsis.”

Lines 125 to 132: I don't understand. How is the reduction of the gametophyte in more advanced land plants connected to the presence of single-celled SAMs in both gametophytes and sporophytes of non-seed plants.

Response: We thank this Reviewer for bringing this to our attention. However, we don’t quite understand the question of the reviewer. Our text might have been confusing because we never meant to suggest that there was a connection between the reduction of the gametophyte in advanced plants and single-cell meristems in non-seed plants. Therefore, we rephrased the text to try to make clearer this point. This part now reads as follows:

“It is remarkable that, as land plants diversified, life cycle generation dominance twisted, and the sporophyte generation became dominant and increasingly specialized, whereas the gametophyte generation became reduced [32,33]. This trend is reflected in SAMs. Non-seed plants have simple meristems, sometimes with a single apical cell, to sustain their growth. In contrast, seed plants' sporophytes develop complex multicellular meristems to support their growth and specialization, while their gametophytes lack meristems [34,35] (Figure 2). However, as both phases have SAMs, it is reasonable to discuss the origin of meristems in terms of the plant life cycle.”

Line 474: What is "a canonical OC"?

Response: We thank this Reviewer for bringing this to our attention. We refer as a canonical OC to those having a WUS-CLV3 regulatory loop. We understand that this may be confusing for the readers; therefore, we edited the text as follows:

“Based on their scRNA-seq data, Satterly and colleagues [128] did not find evidence of a WUS-CLV3 regulatory loop as in other SAM transcriptional landscapes. The Arabidopsis SAM exhibits a repression mechanism of the……”

Minor:

Line 117: Please revise. Not all dotted (or rather dashed) arrows indicate auxin and cytokinin flow in this figure.

Response: The Reviewer is right. We corrected the figure legend as follows to clarify this point:

The dashed arrows beside auxin (AUX) and cytokinin (CK) indicate the hormone flow direction.”

Line 122: Please replace "metistem" with "meristem"

Response: This has been corrected.

Line 127: Please replace semicolon with full stop in this line.

Response: This has been corrected. This section now reads:

“In contrast, seed plants' sporophytes develop complex multicellular meristems to support their growth and specialization, while their gametophytes lack meristems [34,35] (Figure 2). However, as both phases have SAMs, it is reasonable to discuss the origin of meristems in terms of the plant life cycle.”

Line 150: Please correct – "… which suggests the existence of fundamental mechanisms…"

Response: We thank the reviewer for this suggestion that makes our text more precise. The sentence now reads as follows:

“In fact, mosses and ferns gametophyte SAMs share key transcription factors (TFs) with angiosperm sporophyte SAMs [42,43], which suggests the existence of fundamental mechanisms involved in stem cell regulation across land plants.”

Figure 2: This is a beautiful figure; but please reconsider the use of light blue and light pink for the arrows, as these will not visible. Please align the labels at above the box.

Response: Thank you for this great recommendation. As suggested, we modified this figure and now use darker colors for the arrows. The corrected figure has been added to this revised version of the manuscript.

Line 176: Please replace "Tetrahedral" with "tedrahedral"

Response: This has been corrected.

Line 278: Please replace "behind the CZ" with "below the CZ"

Response: The text has been edited accordingly.

Line 317: Please capitalize "Apetala 2"

Response: Apetala 2 has been capitalized.

Line 344: full stop missing after sentence.

Response: This has been corrected.

Line 379: Please replace full stop with comma

Response: This has been corrected and now reads:

“The AP2/ERF family is involved in diverse roles through plant evolution, although there is no experimental evidence of AP2-like function and no ANT sequences in algae. “

Lines 493/494: something is off with this sentence. Please rephrase.

Response:  We thank the Reviewer for this suggestion that makes our text more precise. This sentence has been rephrased and now reads as follows:

“These technologies alleviate the technical difficulties of working on SAMs, as seen in the examples above. However, while there are remarkable advantages to using sc/snRNA-seq for discovering genes and TFs, functional assessment of the regulation dynamics can only be achieved through genetic and genomic approaches. “

Reviewer 2 Report

Comments and Suggestions for Authors

Review of ijms-2649561

The shoot apical meristem: an evolutionary molding of higher plants

Tania Kean-Galeno, Damar Lopez-Arredondo, and Luis Herrera-Estrella

The authors have prepared a very comprehensive and thorough review of our current knowledge about the structure of the shoot apical meristem and its potential evolutionary origin or origins. They describe the meristems in not only angiosperms but also in many other taxonomic groups such as liverworts, mosses, ferns and gymnosperms. They also describe current breakthrough technologies such as SC-seq and sn-seq that are providing significant new information about the SAM. 

The review is well-organized and well-written, with only a few minor issues listed below.

Line 17: Gymnosperms are not a class but a group comprising 4 main phyla. I recommend changing “gymnosperm class” to “gymnosperms”

Line 47: please change “reserach” to “research”

Line 78: I recommend changing “signaling out” to “singling out” and “on totipotency” to “of totipotency

Line 115: change “poll” to “pool”

Line 119: “basipetal” should be “basipetally”

Lines 149-151: please rewrite for clarity and to correct grammatical mistakes.

Lines 155- 157: please rewrite for clarity

Line 173: please clarify “and its division”

Line 174: please change anticlinal to anticlinally and periclinal to periclinally.

Lines 183-185: please rewrite for clarity

Line 213: please explain what a transient cell is.

Line 225: change gametophyte to gametophytic

Lines 274-276: please rewrite for clarity

Lines 320-321: please rewrite for clarity

Lines 378-379 should be a single sentence

Lines 459-463: please rewrite for clarity

Lines 467-473: please rewrite for clarity

Comments on the Quality of English Language

The English is excellent with only a few sentences that are hard to understand.

Author Response

Reviewer 3 

The authors have prepared a very comprehensive and thorough review of our current knowledge about the structure of the shoot apical meristem and its potential evolutionary origin or origins. They describe the meristems in not only angiosperms but also in many other taxonomic groups such as liverworts, mosses, ferns and gymnosperms. They also describe current breakthrough technologies such as SC-seq and sn-seq that are providing significant new information about the SAM. 

The review is well-organized and well-written, with only a few minor issues listed below.

Response: We thank this Reviewer for his/her opinion on our review and the important revisions suggested. The text has been edited accordingly in this revised version of the manuscript, and our responses to each suggestion are provided below.

Line 17: Gymnosperms are not a class but a group comprising 4 main phyla. I recommend changing "gymnosperm class" to "gymnosperms"

Response: This has been corrected.

Line 47: please change "reserach" to "research"

Response: This has been corrected.

Line 78: I recommend changing "signaling out" to "singling out" and "on totipotency" to "of totipotency

Response: Both issues have been corrected.

Line 115: change "poll" to "pool"

Response: This has been corrected.

Line 119: "basipetal" should be "basipetally"

Response: This has been corrected.

Lines 149-151: please rewrite for clarity and to correct grammatical mistakes.

Response: We thank the Reviewer for noticing this. This sentence has been corrected and now reads:

“However, because plants have retained a common set of genetic components since they diverged from algae, it is likely that they also share homologous regulatory loops and mechanisms in regulating their SAM.”

Lines 155- 157: please rewrite for clarity

Response: Thank you for this recommendation. This sentence has rewritten and now reads:

 “Figure 2. Phylogeny of streptophytes and schematic representation of the shoot apical meristem (SAM) location throughout gametophytic and sporophytic life stages.”

Line 173: please clarify "and its division"

Response: We thank the Reviewer for this recommendation. This sentence has been rephrased to improve clarity as follows:

“(a) Representation of meristem shapes and the division plane of meristematic cells.”

Line 174: please change anticlinal to anticlinally and periclinal to periclinally.

Response: Thank you for noticing these issues; both have been corrected.

Lines 183-185: please rewrite for clarity

Response:

“Within bryophytes, a variety of apical meristems are found, including unicellular meristems with one, two, three, and four cutting sides, as well as simple multicellular meristems consisting of two cells with eight cutting sides.”

Line 213: please explain what a transient cell is.

Response: We thank this Reviewer for this recommendation. We meant to refer to a temporary apical cell. Therefore, we edited the text as follows for clarity:

“The moss sporophyte has three types of cells recognized as meristems: the apical cell, active in the early embryo; the central seta cell that maintains the seta's growth; and a single apical cell of the sporophytic tissue [34,61].

Line 225: change gametophyte to gametophytic

Response: This has been corrected.

Lines 274-276: please rewrite for clarity

Response: Thank you for this recommendation. This sentence has been edited as follows:

“SAMs exhibit zonation patterns in gymnosperms and angiosperms, often called cellular domains, which reflect variations in cell division or activity, regardless of the functional layers. “

Lines 320-321: please rewrite for clarity

Response: Thank you for this recommendation. The legend of Figure 4 has been edited to improve clarity as follows:

“Figure 4. SAM signaling pathways in various species. This figure displays genes and factors, such as cytokinin (CK), with established genetic and biochemical interactions. Arrows indicate positive regulation, while barred lines represent negative regulation.”

Lines 378-379 should be a single sentence.

Response: Thank you for this recommendation. This has been corrected:

“The AP2/ERF family is involved in diverse roles through plant evolution, although there is no experimental evidence of AP2-like function and no ANT sequences in algae. Phylogenetic analysis showed that an AP2 domain (AP2-R1 AA insertion) of the microalgae C. reinhardtii and Chlorokybus atmophyticus form a sister clade to the major clade of the AP2-like sequences of plants.”

Lines 459-463: please rewrite for clarity

Response: Thank you for this recommendation. This sentence has been rewritten as suggested and now reads:

“In addition, in-situ hybridization localizes WUS in the SAM. However, in contrast to Arabidopsis, where WUS expression is restricted to the organizing center (OC), in maize, it is detected in peripheral cell layers at the primordium height [135]. It's noteworthy that in maize, SAM sc-RNA-seq failed to detect WUS or any functionally homologous genes [128].”

Lines 467-473: please rewrite for clarity

Response: Thank you for this recommendation. This sentence has been rewritten as suggested and now reads:

“Several challenges deriving from the protoplasting technique could hinder maize SAM heterogeneity due to issues arising from this process. Protoplasting has been linked with the activation of ectopic gene expression, bias on proportions of cell types [8,127], and noise gene expression related to the activation of regeneration and cell division mechanisms after protoplast isolation [96,130]. Therefore, it is possible that WUS-containing cells were affected during protoplast isolation, or the sensitivity of the in-situ sequencing (ISS) could not capture the low level of WUS expression. Considerations on this phenomenon are: the expression of WUS within the maize meristem exhibits variation based on its developmental stage, or there is no WUS-CLV3 regulatory loop in maize SAM under normal conditions. Alternatively, WUS could function as a mobile TF [134], raising the possibility that in the maize SAM, WUS has a different expression pattern than in other plants.”

Reviewer 3 Report

Comments and Suggestions for Authors

The work by Kean-Galeno reviews plant development patterns up-regulated by the shoot apical meristem (SAM), and its phenotypic consequences such as growth of aerial structures, lateral organs and secondary meristems, as well as growth under abiotic stress and its morphological, physiological, and biochemical correlates. The review highlights key findings across the plant kingdom, and envisions targeting further research in gymnosperms. The review is well written and condensed, as well as technically appropriate, besides having excellent artwork. However, before being able to recommend acceptance, I invite authors to address the following amendments.

First, the introduction section should properly close (L47) with an explicit research questions, hypotheses and expected results (so far only a main goal is described) that the review aims exploring. For instance, do authors hypothesize a polygenic or a Mendelian/biomodal segregation for SAM and its phenotypic effects? Are SAM-modulated traits expected to exhibit antagonistic pleiotropy or conditional neutrality when comparing them in various environments?  

Second, methodological speaking and following MDPI’s guidelines to rely on PRISMA procedure (see here http://prisma-statement.org/PRISMAStatement/) to prepare systematic reviews, authors should reported in L64 a specific methodological section explaining the concrete steps/parameters used during the bibliographic search, filtering, and summary of key references. This section can be brief (one paragraph), but still will make the review more systematic.

Third, concerning the literature results and figures, all are very well edited and insightful. Still, authors should try including and explicit quantitative metric to compare among studies, which so far is lacking.

Fourth, when introducing and discussing abiotic stresses in L438, please also comment on potential SAM-based gene functional correlation of nutrient stress tolerance with other stresses such drought, heat, salt and flooding tolerance (consider referring to Front Genet 2019 10:954), which are stresses typically correlated with salt stress in the face of hypoxia and osmotic imbalance (consider referring to PLoS One 2013 8(5):e62898) leading to nutrient stress by accumulation of anti-nutrients. Please comment/discuss the latter list of correlations of SAM-promoted growth with the abiotic responses with DREB-based ABA-independent drought tolerance responses (consider referring to the work Theor Applied Genet 2012 125(5):1069-85, a good place is section in L369 since they belong to the AP2/ERF family type). Any insights applicable to the regulation of?

Fifth, the report is also lacking a very brief paragraph (L495) that describes the major caveats/limitations of the works reviewed so far. It is never beyond the scope of any review to explicitly acknowledge the caveats in the literature, especially when dealing with highly variable trait profiles in a complex trait with strong GxE effects and plasticity such as SAM-promoted growth. For instance, trait variation may exhibit antagonistic pleiotropy or conditional neutrality when comparing their presence and their effects (direction and magnitude) in contrasting environmental treatments, something to be tested in more profound trials beyond a single location). Also, link this point with the perspectives section in L495

Last but not least, a short closing section with conclusions in L533 would be insightful for readers to fill the key take home messages from the review.

Author Response

Reviewer 3

The work by Kean-Galeno reviews plant development patterns up-regulated by the shoot apical meristem (SAM), and its phenotypic consequences such as growth of aerial structures, lateral organs and secondary meristems, as well as growth under abiotic stress and its morphological, physiological, and biochemical correlates. The review highlights key findings across the plant kingdom, and envisions targeting further research in gymnosperms. The review is well written and condensed, as well as technically appropriate, besides having excellent artwork. However, before being able to recommend acceptance, I invite authors to address the following amendments.

First, the introduction section should properly close (L47) with an explicit research questions, hypotheses and expected results (so far only a main goal is described) that the review aims exploring. For instance, do authors hypothesize a polygenic or a Mendelian/biomodal segregation for SAM and its phenotypic effects? Are SAM-modulated traits expected to exhibit antagonistic pleiotropy or conditional neutrality when comparing them in various environments?  

Response: We thank this reviewer for pointing out the need to include research questions at the end of the introduction. We also thank the reviewer for suggesting these two very interesting questions regarding SAM regulation and associated traits, which are important to consider. However, the suggested questions are briefly addressed in the current version of this review article. Therefore, we have incorporated two questions that are central to the main objective of this review: 1) Do plant meristems have a single origin or multiple independent events gave rise to plant meristems? and 2) Are the circuits regulating the maintenance of stem cells conserved during plant evolution?

Second, methodological speaking and following MDPI's guidelines to rely on PRISMA procedure (see here http://prisma-statement.org/PRISMAStatement/) to prepare systematic reviews, authors should reported in L64 a specific methodological section explaining the concrete steps/parameters used during the bibliographic search, filtering, and summary of key references. This section can be brief (one paragraph), but still will make the review more systematic.

Response: We thank the Reviewer for this suggestion. We understand that systematic reviews and meta-analyses must follow Prisma procedures, but these do not apply to non-systematic reviews, as is our case (https://www.mdpi.com/about/article_types). In this review, we are not trying to analyze quantitative data but rather integrate heterogeneous information to provide a general overview of plant meristems' state of the art. As information comes from heterogeneous studies and many different techniques, specifying standards to filter and systematize the information is impossible. Nevertheless, we included the following paragraph:

“This review was prepared by searching all bibliographies on plant meristems in PubMed and Google Scholar databases. The search was filtered using keywords including shoot apical meristem, root apical meristem, plant stem cells, evolution of plant meristems, and single-cell analysis of plant meristems. When multiple papers were found on the same topic, the most complete and updated reports with the most citations, were selected.”

Third, concerning the literature results and figures, all are very well edited and insightful. Still, authors should try including and explicit quantitative metric to compare among studies, which so far is lacking.

Response: We thank the Reviewer for pointing this out. Most of the data described and discussed in this review are of a qualitative nature, and no quantitative data insight can be added. The only quantitative data is those related to single-cell RNA-seq experiments. Since the data is from different species, quantitative comparisons are hard to interpret. Nevertheless, we prepared the following table to compare the multiple SAM sn/sc-RNA-seq research papers. This table has been added to this revised version of the manuscript.

Table 1. Summary of snRNA-seq and snRNA-seq SAM studies regarding SAM.

Scientific name

Clade

Technology

Validation

Focus

Reference

Arabidopsis thaliana

Dicot

scRNA-seq

GFP-reporter lines of identified genes (At2g38300, At4g34970 (ADF9), At4g11290, and At1g13650).

Construction of developmental trajectories of several tissues and Integrative.

[127]

Zea mays

Monocot

scRNA-seq

In situ hybridization of KN1-overexpressor lines.

Analysis of maize SAM stem-cell function and cell-fate acquisition.

[128]

Pisum sativum

Dicot

scRNA-seq

Laser capture microdissection to identify marker genes.

Cell-type-specific responses to Boron deficiency.

[129]

Solanum lycopersicum

Dicot

snRNA-seq

Homologous markers identification.

Construction of developmental trajectories of several tissues.

[130]

Populus trichocarpa

Dicot

snRNA-seq

RNA in situ hybridization and GUS-reporter lines.

Comparisons on Phloem and xylem development between a woody plant and annual herb.

[131]

Fourth, when introducing and discussing abiotic stresses in L438, please also comment on potential SAM-based gene functional correlation of nutrient stress tolerance with other stresses such drought, heat, salt and flooding tolerance (consider referring to Front Genet 2019 10:954), which are stresses typically correlated with salt stress in the face of hypoxia and osmotic imbalance (consider referring to PLoS One 2013 8(5):e62898) leading to nutrient stress by accumulation of anti-nutrients. Please comment/discuss the latter list of correlations of SAM-promoted growth with the abiotic responses with DREB-based ABA-independent drought tolerance responses (consider referring to the work Theor Applied Genet 2012 125(5):1069-85, a good place is section in L369 since they belong to the AP2/ERF family type). Any insights applicable to the regulation of?

Response: We thank the reviewer for this recommendation. Although the effect of environmental factors on the function of plant meristems is quite important, we consider this topic to be out of the scope of this review. Nevertheless, we agree with the Reviewer that the effect of abiotic stress should be at least mentioned in our review. Therefore, we added the following paragraph that includes several of the publications suggested by the reviewer:

New Line 949:

SAM is typically studied under standard growth conditions. However, environmental stresses can significantly impact plant growth, highlighting the crucial link between SAM and stress responses. Investigating this connection may reveal strategies that allow plants to adapt to adverse environments. An interesting perspective, following the sc-RNA-seq study on pea [129], is to extend the research to other legumes, such as common bean (Phaseolus vulgaris), an important crop that is known to be susceptible to drought stress [141]. By employing sn/sc-RNA-seq to analyze SAM bean under drought stress at a cellular level, it is possible to explore the involvement of key TFs in its drought stress response; for example, DREB2 (Dehydration-Responsive Element-Binding Protein 2), a member of the AP2/ERF family, which has been described to play a fundamental role in drought responses [142].”

Fifth, the report is also lacking a very brief paragraph (L495) that describes the major caveats/limitations of the works reviewed so far. It is never beyond the scope of any review to explicitly acknowledge the caveats in the literature, especially when dealing with highly variable trait profiles in a complex trait with strong GxE effects and plasticity such as SAM-promoted growth. For instance, trait variation may exhibit antagonistic pleiotropy or conditional neutrality when comparing their presence and their effects (direction and magnitude) in contrasting environmental treatments, something to be tested in more profound trials beyond a single location). Also, link this point with the perspectives section in L49.  Last but not least, a short closing section with conclusions in L533 would be insightful for readers to fill the key take home messages from the review.

Response: We thank this Reviewer for bringing this to our attention. As recommended, in this revised version of the manuscript, we added the following paragraphs to the Perspectives section, which now read as follows:

New Line 890:

We have embarked on the synthesis of SAM from an evolutionary perspective. However, broadening our view to encompass an evolutionary perspective has inherent limitations. While this approach has been a subject of discussion for some time, early research was restricted by microscopy techniques, resulting in a substantial body of literature that primarily focused on descriptive studies and qualitative data, often employing non-standardized characters for comparison. Nevertheless, refined microscopy techniques are now available and facilitate the morphological comparison of meristems between species [24,70]. In general, techniques for studying meristems at various levels have only recently been developed. In addition to these limitations associated with evolutionary perspectives, the lack of models spanning phylogeny is notable. Consequently, in silico, in situ, and in vitro studies are only sometimes available for particular clades, creating significant gaps in our understanding of the evolutionary processes.

New Line 965:

“Another exciting perspective is the study of stress responses at the SAM level to explain early responses to anatomical alterations and to search for conserved mechanisms that promote and maintain cell dedifferentiation and proliferation, which would greatly impact diverse research fields.

New Line 970:

With the advent of new technological alternatives, it is now possible to trace the conservation of regulatory pathways and loops, i.e., those involved in stem cell proliferation and maintenance. Additionally, these technologies enable the identification of key genes responsible for specific traits. However, technologies like sc/snRNA-seq analysis are recent in the plant field. Their application in uncovering key genes on stress response has begun to shed light on potential applications in the crop industry and plant improvement.

Round 2

Reviewer 3 Report

Comments and Suggestions for Authors

Authors have done a great work improving the review. I would simply recommend to do not forget mentioning the PRISMA methodology (http://prisma-statement.org/PRISMAStatement/default.aspx?AspxAutoDetectCookieSupport=1) for the literature search. That is something that could easily be included at the production step. 

Author Response

Dear Editor,
I am attaching the revised version of our manuscript by Kean-Galeno et al. that we submitted for consideration to the International Journal of Molecular Sciences. We prepared this revised version considering all the comments from the reviewers and included the additional references suggested.

Although the reviewer had only minor modifications, we are grateful to this reviewer because it helped us to clarify several concepts and improved the clarity and quality of our review.

Sincerely

Luis Herrera estrella

Dear Lev-Yadun,

We are grateful for your revision on botanical aspects. As you mentioned, there were misconceptions that we corrected in our revised manuscript. Joining molecular and structural aspects was challenging, but your revision was extremely helpful. We have addressed all the comments you provided. We hope you agree with the changes we made. You will find comments organized following the section they belong in the manuscript, and behind each comment, we show the corrected version.

(1) From many aspects it is a very good review and I like it, but like most if not all excellent molecular scientists, they have holes in their botany (the very good molecular geneticist that thought my son “introduction to botany” two years ago, thought the class pure hallucinations in shoot development), expressed mostly in the introduction. I will do my best to fix these holes because otherwise this important and review may mislead the young generation.

(2) Line 12. Replace secondary with primary. Secondary meristems are either not formed at all (e.g., in Cereals) or formed later and far from the shoot apex and not from the shoot apex!

Response: done

Line 13: Abstract: shoot apical meristem (SAM) gives rise to the aerial structure of plants by producing lateral organs and other meristems.

(3) Line 33. I refer to “Additionally”. In very many cases, in gymnosperms, dicots and monocots, the plants produce adventitious apices (both shoot and root) sometimes by the millions!  They are formed from several cell types in shoots, roots, and even leaves, and from cells that ceased to express cell cycle for various periods of time, even for years. THEREFORE, SHOOT APICES ARE ALWAYS DIFFERENTIATED!  Moreover, there are no non-differentiated cells in plants!!!!!!! There are non-differentiated cells when we produce protoplasts, but this is an in-vitro system and not in-vivo!  I wrote about it 20 years ago (Lev-Yadun, S. 2003). Stem cells in plants are differentiated too. Current Topics in Plant Biology 4:93-102) (attached) from a developmental point of view.  Later, a decade later or so, Eliott Meyerowitz wrote about it from a molecular point of view, but I do not remember the reference.

Response: done

Line 35: Additionally, plants have lateral meristems, which are responsible for secondary growth that determines the thickness of the plant, intercalary meristems, which are responsible for growth at the base of the nodes and leaf blades mainly in monocots, and buds (classified as to location contents or activity) which usually contains a small mass of meristematic tissue in spermatophytes [1–3].

(5) Lines 75, 106. “undifferentiated” see comment # 3.

(7) Lines 79-80, 87-88. No cell including the zygote is “totipotent”.  Cells can be pluripotent or not potent at all if they have a thick secondary cell wall, or if like sieve elements they have lost their nucleus.

Response: we restructured this section considering your corrections and added the references you recommended; the following is the revised version of the corresponding section.

Line 79: Plant stem cells are typically found in meristems, and their defining characteristics include: being the least differentiated cells, self-maintenance, slow proliferation with an extended G1 (or G0) cell state, pluripotency (ability to differentiate into multiple cell types), and regenerative potential or the ability to repopulate after damage [6–8]. The study of apical meristems (AMs) is particularly intriguing due to their stem cell pool, also known as promeristems; located at the central part of the shoot and the root apex. A continuous differentiating cell flux occurs throughout plant development from stem cells to proliferating cells, finishing with differentiating cell states [9]. Two types of plant cell differentiation exist: terminal and non-terminal. Terminal differentiation implies that a cell can no longer change its fate, while non-terminal differentiation allows a cell to change its fate when exposed to the correct signals, such as normal ontogeny, wounding, or other physiological changes [10]. Cells undergoing non-terminal differentiation appear to retain a stem cell potential [3]. However, the specific conditions activating their differentiation into various cell types remain largely unknown [11]. Standard models for studying plant cell differentiation include embryos, protoplasts, calli, and meristems [11–14]. These models are valuable as they contain pluripotent or totipotent cells, easily different, ultimately giving rise to different plant tissues. Hence, an interesting perspective to understand this phenomenon is to focus on the evolution of pluripotency by comparing the meristems of early and modern diverged land plants, signaling out the common factors of pluripotency through evolution.

(8) Line 93. The shoot and root apices CANNOT be considered as primary meristems!!!!!!!!  The primary meristems in the shoot are: (1) Protoderm, (2) ground meristem, (3) procambium.  In the root there is also the calyptrogen.  Sometimes, in the root, some of them may be referred to by different names.

(9) Line 94.  While the self-renewing stem cells are indeed determined during embryogenesis, they can be formed adventitiously from mature tissues or following wounding.

Response: We agree with your suggestion and added the concept of promeristems.

Line 100: Extensive research has been conducted to understand RAM evolution, but little is known about the SAM evolutionary process. Since both AMs are structures with similar characteristics, studying the RAM can be a starting point for understanding the SAM. Some characteristics are that they both are reservoirs of stem cells, they are promeristems, that give rise to five primary meristems (protoderm, ground meristem, procambium, peri-cycle and calyptrogen in the root [10], and they are determined during embryogenesis [15]. Stem cells in the root apical meristem (RAM) are present and maintained in the quiescent center (QC). In contrast, in the shoot apical meristem (SAM), stem cells are contained in the Central Zone (CZ) and maintained by the organizing center (OC) [16]. The QC and OC are the signaling centers responsible for stem cell maintenance in both meristems [16] and have been proposed to be functionally equivalent [17,18].

(10) Line 116, add “that” after “revealed.

Response: done

Line 128: Genomic analyses revealed limited gene expansion occurred at the divergence between the lycophyte and euphyllophyte clades [31].

(11) Lines 147, 151, 154, 155, 226.  I suggest to change “theory” to “hypothesis”.

Response: done

(12) Line 171  It should be “In non-seed vascular plants”.  In seed plants indeed only the sporophytes have meristems, but non-seed vascular plants may have large gametophytes that grow and have meristems.

(13) Lines 171-172. The say “The gametophytes of lycophytes and spermatophytes lack meristems” is totally wrong.  Some of them, especially female ones are large and have meristems (see Gifford, E.M., Foster, A.S. 1989. Morphology and evolution of vascular plants. 3rd ed. W. H. Freeman and Company, New York).

Response: We consider your comment, make the proper corrections, and add a paragraph explaining which meristems we focus on and why.

Line 184: Figure 2. Phylogeny of streptophytes and schematic representation of the shoot apical meristem (SAM) location throughout gametophytic and sporophytic life stages. The pink arrows point to the position of the meristems, while the blue arrows indicate a close-up of specific meristematic structures. Within bryophytes and charophytes, the gametophyte is the dominant life state with vegetative growth from SAMs. While in vascular plants, vegetative growth from SAMs occurs in a sporophyte state. The gametophytes of lycophytes and spermatophytes are not presented.

Line 158: However, we must mention that while we are focusing on SAM, specifying apical meristems of the vegetative growth (independent of the presence of leaves). We include some meristems without or with reduced vegetative growth (as in bryophytes) [40] as they are attractive for hypotheses on the evolution of SAM

(14) All over the text.  Names of genera such as Arabidopsis or Populus should be in italics.

Response: done

(15) Lines 284-287.  Cells from L1, L2, L3 are known from at least many decades if not for a century to switch between them as can be seen in variegated chimeras (lots of references) and also in longitudinal sections in shoot apical meristems of various plants.  Therefore, the say in lines 286-287 “It has been suggested that each layer comprises its own group of initials” is a clear mistake.  It should be deleted with reference 77 there and anywhere where reference 77 appears.

Response: done

Line 297: SAMs in seed plants tend to be larger and more complex compared to other land plants. They consist of multiple initial cells and distinct functional layers. The outer layer (L1), also known as tunica, gives rise to the epidermis; and L2 in maize, or L2 and L3 (corpus) in Arabidopsis are the internal layers [88–90].

(16) Lines 430-431.  The say that it (apical dominance) has not been validated in perennial plants is not true.  Kozlowski, T.T. (1971) Vol. 1. (Growth and development of trees. Academic Press) discussed it a lot.  For instance, pruning of tree tops shows it in both conifers and dicots.  I simply have no time to expand on this.  It should be deleted with the relevant references.  Moreover, strigolactones are involved, probably more than auxin, and the authors should deal with it in that section.

Response: We searched for your recommended references and made the correct amendment. Additionally, we add a paragraph introducing the participation of strigolactones in apical dominance.

Line 552: the phytohormone auxin produced by the shoot tip is transported in a basipetal fashion by the polar transport stream, which inhibits axillary bud outgrowth [92,95]. Although this concept has been observed in angiosperms and gymnosperms [2,96–98], it has not been validated in other growth systems as dichotomous branching.

Line 606: The apical dominance mechanism explains the classical observations; however, in addition to auxins and cytokinins, strigolactones (SLs) constitute a new class of phytohormones related to this mechanism. SLs were not considered in classic literature because they were not introduced until the 2000s [101,102] and were soon linked to multiple developmental processes, such as shoot development [103–106]. Specifically, SLs play a crucial role in repressing bud outgrowth in monocots and in responses to environmental factors [104,107,108]. Recent studies suggest that the inhibition of bud outgrowth in apical dominance is attributed to the modulation of apically derived auxin flux by cytokinin and SLs [103,104].